# Deep learning for the prediction of early on-treatment response in metastatic colorectal cancer from serial medical imaging

Lin Lu [1], Laurent Dercle[1], Binsheng Zhao[1] & Lawrence H. Schwartz [1✉]

In current clinical practice, tumor response assessment is usually based on tumor size change on serial computerized tomography (CT) scan images. However, evaluation of tumor response to anti-vascular endothelial growth factor therapies in metastatic colorectal cancer (mCRC) is limited because morphological change in tumor may occur earlier than tumor size change. Here we present an analysis utilizing a deep learning (DL) network to characterize tumor morphological change for response assessment in mCRC patients. We retrospectively analyzed 1,028 mCRC patients who were prospectively included in the VELOUR trial (NCT00561470). We found that DL network was able to predict early on-treatment response in mCRC and showed better performance than its size-based counterpart with C-Index: 0.649 (95% CI: 0.619,0.679) vs. 0.627 (95% CI: 0.567,0.638), $p = 0.009$, $z$-test. The integration of DL network with size-based methodology could further improve the prediction performance to C-Index: 0.694 (95% CI: 0.661,0.720), which was superior to size/DL-based-only models (all $p < 0.001$, $z$-test). Our study suggests that DL network could provide a noninvasive mean for quantitative and comprehensive characterization of tumor morphological change, which may potentially benefit personalized early on-treatment decision making.

[1] Department of Radiology, Columbia University Irving Medical Center, New York, NY 10032, USA. ✉email: lhs2120@cumc.columbia.edu

 1

n clinical trials of systemic therapies for solid tumors, response is often measured by imaging endpoints such as progression-free survival (PFS) and response rate (RR). In solid tumors, these endpoints are based on Response Evaluation Criteria In Solid Tumours (RECIST) 1.1 criteria[1,2] and assessed using serial computerized tomography (CT) images. Progression is defined as an increase from baseline of 20% or more in the longest diameter of target lesions, while response is similarly defined as decrease of 30% or more.

There is a desire for criteria which can assess response more optimally and provide better guidance for clinical decision in optimal treatment of metastatic colorectal cancer (mCRC) patients. Early tumor shrinkage (ETS), defined as decrease in tumor load measured at the time of first restaging, has recently come to the attention of clinicians as a promising predictive early on-treatment imaging biomarker for long-term outcome (overall survival, OS) in mCRC patients[3]. Retrospective analysis of clinical trials has shown that ETS criteria can identify prolonged survivors who show clear benefit from continuation of treatment, especially in first-line clinical trials which compared anti-EGFR agent plus chemotherapy vs. chemotherapy alone[4].

While both ETS and RECIST criteria rely on changes in tumor size for their assessment of response, other morphological changes, such as changes of overall attenuation, tumor–tissue interface and peripheral rim of enhancement[5], may occur even earlier in treatment than tumor size change. For instance, in most anti-VEGF therapies, changes in tumor architecture sometimes happen prior to tumor shrinkage[5–7]. Many qualitative and quantitative image analysis algorithms for assessing tumor morphological change have been proposed over the last decade[8–11]. These medical image analysis algorithms, also known as radiomics approaches[12], allow high-throughput conversion of medical images into mineable quantitative data, which make it feasible to assess tumor morphological changes quantitatively. For example, a recent study proposed by Dohan et al.[9] used a radiomics signature that included three radiomics features to predict treatment

outcome for unresectable hepatic metastases in patients with colorectal cancer. However, these medical image analysis algorithms are usually subject to human predefined criteria, typically involving manual or semi-automatic segmentation of the region of interest (i.e., the tumor region) and using a priori human-engineered image features.

In recent years, medical image analysis has been increasingly shifting towards the deep learning (DL) method[13], which has demonstrated remarkable results in a variety of medical image application, including dermatology[14], ophthalmology[15], pathology[16], and radiology[17,18]. The DL method were data-driven approaches in which image features are automatically designed and organized based on features' predictive ability instead of human pre-knowledge. However, until now, there is seldom report on applying DL method on predicting early on-treatment response in oncology, especially mCRC. For example, when searching in the National Library of Medicine medical literature database with the key words of 'deep learning', 'metastatic colorectal cancer' and 'response or survival' (https://pubmed.ncbi.nlm.nih.gov/), only one literature was found[19] and the literature was only about colorectal liver metastases with small data. The challenges of predicting mCRC treatment response by using DL method include, (1) mCRC involves multiple metastatic lesions involving multiple organs, most commonly, liver, lung and lymph nodes; (2) the prediction of response involves CT images of multiple time points while the patient is on treatment rather than a single time point; and (3) DL method requires large dataset for training.

Therefore, the aim of this proof-of-concept study was to explore the ability of DL method to predict the early on-treatment response in mCRC patients by using OS as the primary end point. To address the challenges mentioned above, a total of 1028 mCRC patients collected from the VELOUR trial (an international prospective multi-institutional study[20]) were used for our study (see Fig. 1), and a sophisticated DL network architecture, which combines the convolutional neural network (CNN)[13] and

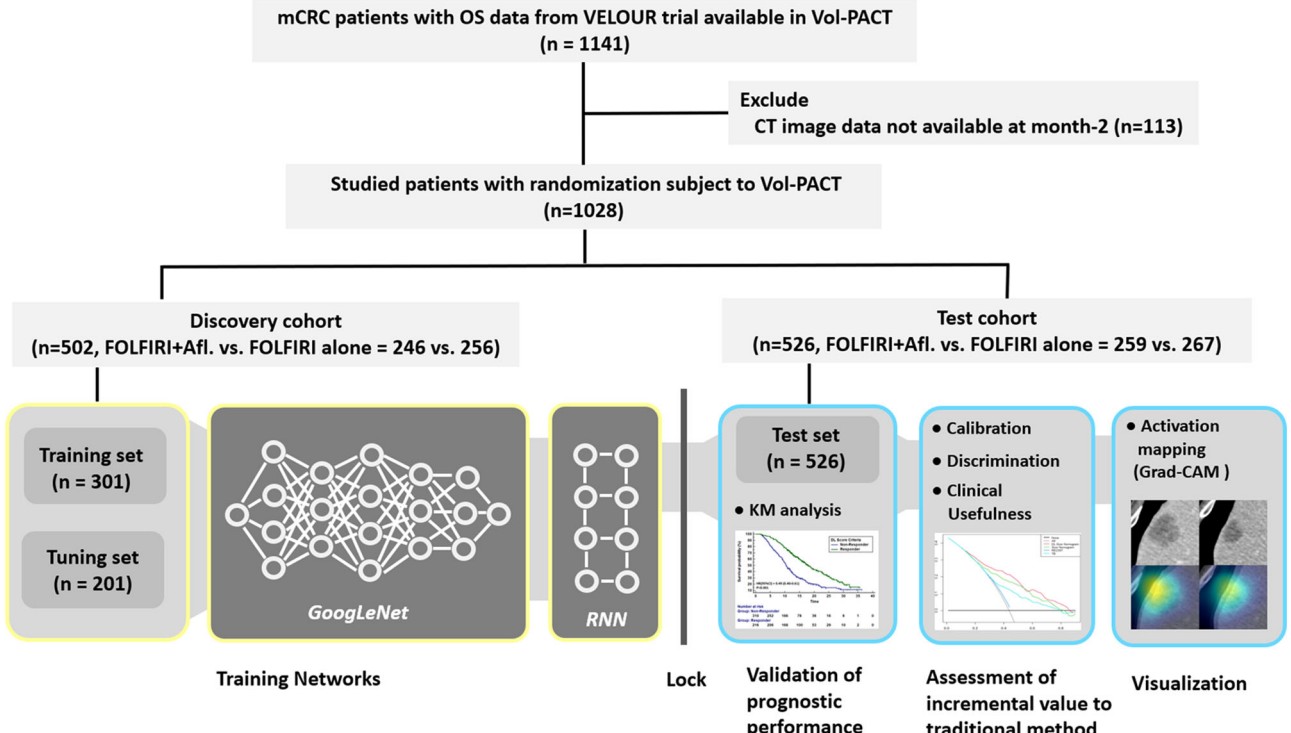

**Fig. 1** Overview of study design.

the recurrent neural network (RNN)[21], were proposed (see Fig. 2). The CNN was used to characterize lesions from different organs, while the RNN was used to characterize the lesions at multiple time points. Our work builds upon a preliminary study published in[22], in which DL networks were utilized as a computational tool to predict lung cancer treatment response. In this study, of a different tumor type, colorectal cancer, a more sophisticated DL networks and larger data were used, and the performance of DL-based method as well as its incremental value to traditional size-based method were assessed. In addition, to gain a better understanding of the DL-based-response radiological phenotypes, we mapped network-attention regions in images as per their contributions to the prediction of OS

by using the gradient-weighted activation mapping method (Grad-CAM)[23].

## Results

**Patient characteristics.** A total of 1028 patients derived from the VELOUR trail were included in this study. Details for patient characteristics of VELOUR trail can be found in existing literatures[20,24]. The 1028 patients were randomized and assigned into discovery ($n = 502$ patients) and test ($n = 526$ patients) cohorts. The media OS for patients in the discovery and test cohorts were 13.56 and 12.71 months, respectively ($p = 0.691$, log-rank test) (see Supplementary Fig. 1). The two cohorts involved 3757 lesions (averagely $3.53 \pm 2.19$ target lesions per patient) coming from a total of 14 anatomic sites (see Table 1). And the main lesion locations were liver (58.3%), lung (22.5%), and lymph node (12.0%). There was no significant difference between the discovery and test cohorts in terms of average number, size, and anatomic position on target lesion (all $p > 0.05$, $t$-test for numeric data and chi-square test for categorical data).

On the aspect of CT scanning, since the VELOUR trail was an international multi-institutional study, the CT images were of great heterogeneity in imaging acquisition settings (see Table 2). As shown in Table 2, there were a total of 3193 CT scans used, deriving from a total of 9 CT manufacturers, 81 manufacturer models, and 65 CT image reconstruction algorithms, respectively. There was a significant difference between the discovery and test cohorts in terms of manufacturer models and CT image reconstruction algorithms (both $p < 0.05$, chi-square test). For those regular imaging parameters, e.g., slice thickness, voltage, product of tube current and time, and pixel spacing, there was no significant difference (all $p > 0.05$, $t$-test, except the product of tube current and time) and mainstream ranges of settings were covered.

**Validation of prognostic performance for DL prediction score.** The concordance correlation coefficient (CCC)[25] for DL prediction score was 0.901 (95% CI: 0.880,0.913), suggesting a good reproducibility of prediction score under potential variation of ROI selection. The AUC (95% CI) of DL prediction score on the tuning set to classify DL-responder/ non-responder was 0.76 (95% CI: 0.72,0.80), and the optimal stratification cutoff was 'DL prediction score $\geq 0.6$' according to the Youden-Index[26] (see

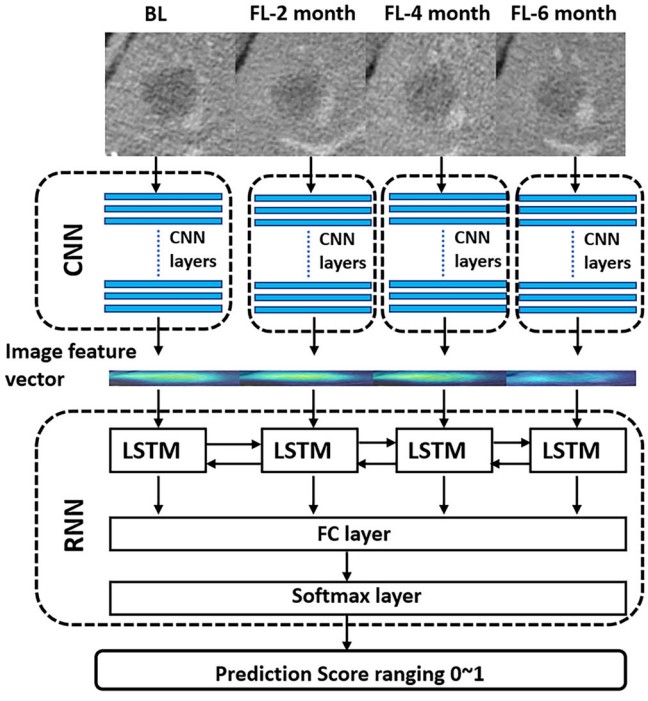

**Fig. 2** The architecture of the proposed DL network.

**Table 1 Lesion characteristics.**

|  | All lesions | Train | Test | p |
|---|---|---|---|---|
| Total number | 3757 | 1864 | 1893 |  |
| Number of selected lesions per patient (mean (±SD)) | 3.53 (2.19) | 3.53 (2.23) | 3.54 (2.15) | 0.952 |
| Lesion size (mm, mean (±SD)) | 29.72 (±21.79) | 29.99 (±19.78) | 30.39 (±20.03) | 0.546 |
| Anatomic position (n (%)) |  |  |  | 0.122 |
| Abdomen | 119 (3.2) | 55 (3.0) | 64 (3.4) |  |
| Adrenal | 57 (1.5) | 33 (1.8) | 24 (1.3) |  |
| Bone | 1 (0.02) | 0 (0.0) | 1 (0.1) |  |
| Kidney | 3 (0.06) | 1 (0.1) | 2 (0.1) |  |
| Liver | 2190 (58.3) | 1087 (58.3) | 1103 (58.3) |  |
| Lung | 847 (22.5) | 427 (22.9) | 420 (22.2) |  |
| Lymph node | 449 (12.0) | 216 (11.6) | 233 (12.3) |  |
| Ovary | 3 (0.06) | 2 (0.1) | 1 (0.1) |  |
| Pancreas | 3 (0.06) | 1 (0.1) | 2 (0.1) |  |
| Pelvis | 23 (0.6) | 17 (0.9) | 6 (0.3) |  |
| Pleural | 3 (0.06) | 3 (0.2) | 0 (0.0) |  |
| Soft tissue | 45 (1.2) | 15 (0.8) | 30 (1.6) |  |
| Spleen | 13 (0.8) | 7 (0.4) | 6 (0.3) |  |
| Thyroid | 1 (0.02) | 0 (0.0) | 1 (0.1) |  |

Note: p-Values were calculated via t-test and chi-square test for numerical and categorical data, respectively. All tests are two sided.

**Table 2 CT scanning characteristics.**

| | All CT scans | Train cohort | Test cohort | p |
|---|---|---|---|---|
| Number of scans | 3193 | 2141 | 1052 | |
| Manufacturer (n (%)) | | | | 0.26 |
| GE MEDICAL SYSTEMS | 1059 (33.2) | 694 (32.4) | 365 (34.7) | |
| Philips | 445 (13.9) | 315 (14.7) | 130 (12.4) | |
| SIEMENS | 1314 (41.2) | 889 (41.5) | 425 (40.4) | |
| TOSHIBA | 312 (9.8) | 204 (9.5) | 108 (10.3) | |
| Five other manufacturers | 63 (2.0) | 39 (1.8) | 24 (2.3) | |
| Manufacturer models (n (%)) | | | | <0.05 |
| Aquilion | 243 (7.6) | 157 (7.3) | 86 (8.2) | |
| Brilliance 64 | 166 (5.2) | 114 (5.3) | 52 (4.9) | |
| Definition | 161 (5.0) | 124 (5.8) | 37 (3.5) | |
| HiSpeed NX/i | 132 (4.1) | 104 (4.9) | 28 (2.7) | |
| LightSpeed VCT | 240 (7.5) | 136 (6.4) | 104 (9.9) | |
| LightSpeed16 | 197 (6.2) | 128 (6.0) | 69 (6.6) | |
| Sensation 16 | 303 (9.5) | 196 (9.2) | 107 (10.2) | |
| Sensation 64 | 309 (9.7) | 216 (10.1) | 93 (8.8) | |
| Seventy-three other models | 1442 (45.2) | 966 (45.1) | 476 (45.2) | |
| CT image reconstruction algorithms (n (%)) | | | | <0.05 |
| B | 249 (7.8) | 165 (7.7) | 84 (8.0) | |
| B30f | 332 (10.4) | 226 (10.6) | 106 (10.1) | |
| B31f | 233 (7.3) | 172 (8.0) | 61 (5.8) | |
| LUNG | 118 (3.7) | 82 (3.8) | 36 (3.4) | |
| STANDARD | 623 (19.5) | 399 (18.6) | 224 (21.3) | |
| STD+ | 141 (4.4) | 108 (5.0) | 33 (3.1) | |
| Fifty-nine other algorithms | 1497 (46.9) | 989 (31.0) | 508 (15.9) | |
| Slice thickness (n (%)) | | | | 0.301 |
| ≤3 mm | 531 (16.6) | 340 (15.9) | 191 (18.2) | |
| 3–5 mm | 2335 (73.1) | 1582 (73.9) | 753 (71.6) | |
| >5 mm | 327 (10.2) | 219 (10.2) | 108 (10.3) | |
| Voltage (n (%)) | | | | 0.52 |
| 120 kVp | 2751 (86.2) | 1841 (85.9) | 910 (86.5) | |
| Others | 442 (13.8) | 300 (14.0) | 142 (13.5) | |
| Product of tube current and time (n (%)) | | | | <0.05 |
| ≤100 mAs | 654 (20.5) | 449 (21.0) | 205 (19.5) | |
| 100~200 mAs | 1196 (37.5) | 782 (36.5) | 414 (39.4) | |
| 200~400 mAs | 757 (23.7) | 515 (24.0) | 242 (23.0) | |
| >400 mAs | 261 (8.2) | 70 (3.3) | 191 (18.2) | |
| NAN | 325 (10.2) | 325 (15.2) | 0 (0.0) | |
| Pixel spacing (n (%)) | | | | 0.408 |
| 0.5~0.75 mm | 1528 (47.9) | 1010 (47.2) | 518 (49.2) | |
| 0.75~1.00 mm | 1640 (51.4) | 1112 (51.9) | 528 (50.2) | |
| >1 mm | 25 (0.8) | 19 (0.9) | 6 (0.6) | |

Note: p-Values were calculated via two-sided chi-square test.

Supplementary Fig. 2). When applying the 'DL prediction score ≥ 0.6' to stratify patients into groups of DL-responder and DL-non-responder in the test cohort, the DL-responders had a significant better OS than the DL-non-responders, with median OS 18.0 vs.10.4 months, hazard ratio (HR) (95% CI) = 0.49 (0.40,0.61), $p = 1 \times 10^{-6}$, log-rank test (see Fig. 3a). A landmark analysis with similar results is provided as Supplementary Fig. 3.

As a comparison, the size-based RECIST criteria, which defines response as decrease of tumor burden of 30% or more, resulted in median OS 17.2 vs.12.5 months, HR (95% CI) = 0.60 (0.42,0.85), $p = 0.02$, log-rank test (see Fig. 3b). The percentage of responders defined by DL criteria vs. RECIST criteria was 41% vs. 6.7%. When compared to another size-based criterion, the ETS criterion (the optimal stratification cutoff for the ETS criterion was 'ETS ≥ 5%' (see Supplementary Fig. 4), DL network still showed better performance in terms of a larger delta median OS, with 7.3 vs. 5.4 (Δ month) and 7.8 vs. 4.6 (Δ month) in the FA and F arms, respectively. Linear correlation was performed to test the association between DL score and ETS (see Supplementary Fig. 5). It was found that there was a weak linear correlation between the DL prediction score and ETS with correlation coefficient (95% CI) = 0.45 (0.38,0.52) and $R^2 = 0.21$. A comparison and supplementary discussion between DL-based and size-based criteria that are based on existing literatures were also provided in the Supplementary Table 1.

**Assessment of incremental value of DL-based method to traditional size-based method.** As shown in Table 3, seven prognostic models were built, i.e., the RECIST, TB, ETS, DL-BS, DL-PS, Size-Nomo, and DL-Nomo models. The DL-Nomo model as well as its calibration curve[27] on 1-year survival prediction is presented in Fig. 4 (nomograms for the other six models and their calibration curves were provided in Supplementary Figs. 6–8). In Fig. 4, we could see that the DL-Nomo model showed good agreement between the prediction of model and the actual outcome.

The Harrell concordance index (C-Index)[28] for the seven prognostic models are listed in Table 4. On the aspect of discrimination performance of models in test cohort, some important observations could be attained. Firstly, the performance of DL-PS model was better than that of its counterpart, the ETS model (0.649 (95% CI: 0.619,0.679) vs. 0.627 (95% CI: 0.567,0.638), $p = 0.009$, z-test), which again indicated the superior of DL-PS to ETS; secondly, the performance of DL-PS model was better than that of the DL-BS model (0.649 (95% CI: 0.619,0.679) vs. 0.607 (95% CI: 0.574,0.639), $p = 8.6 \times 10^{-5}$, z-test), showing the necessity of using multiple time points rather than only using baseline; thirdly and the most importantly, the DL-Nomo model, which incorporated DL-PS into the Size-Nomo model, achieved the best performance among all the models (0.694 (95% CI: 0.661,0.720), all $p < 1 \times 10^{-3}$, z-test), suggesting improvement of integration of DL-based method to traditional size-based method.

Further, a decision curve analysis[29] showed that the multivariate DL-Nomo model had a higher overall net benefit than all other univariate models as well as the multivariate Size-Nomo model across the majority of the range of reasonable threshold probabilities in both of the discovery and test cohort (see Fig. 5). Another interesting finding was that the RECIST model added little clinical value to the early prediction of OS, although it showed acceptable discrimination performance (0.657 (95% CI: 0.546,0.769)). This is because, the RECIST criteria which defined response as decrease of tumor burden of 30% or more, could only definitively identify a limited number of patients in the early treatment period (see Fig. 3b).

**Activation mapping of network.** The magnitude of intensity in the activation mapping indicates the 'importance' of each pixel in CT image that contribute to the eventual prediction. Activation mapping may help clinicians gain understanding of regions within CT images where the predictions of DL network were derived, and thus provide more information inside/outside tumor region. Examples for four patients are presented in Fig. 6, all of

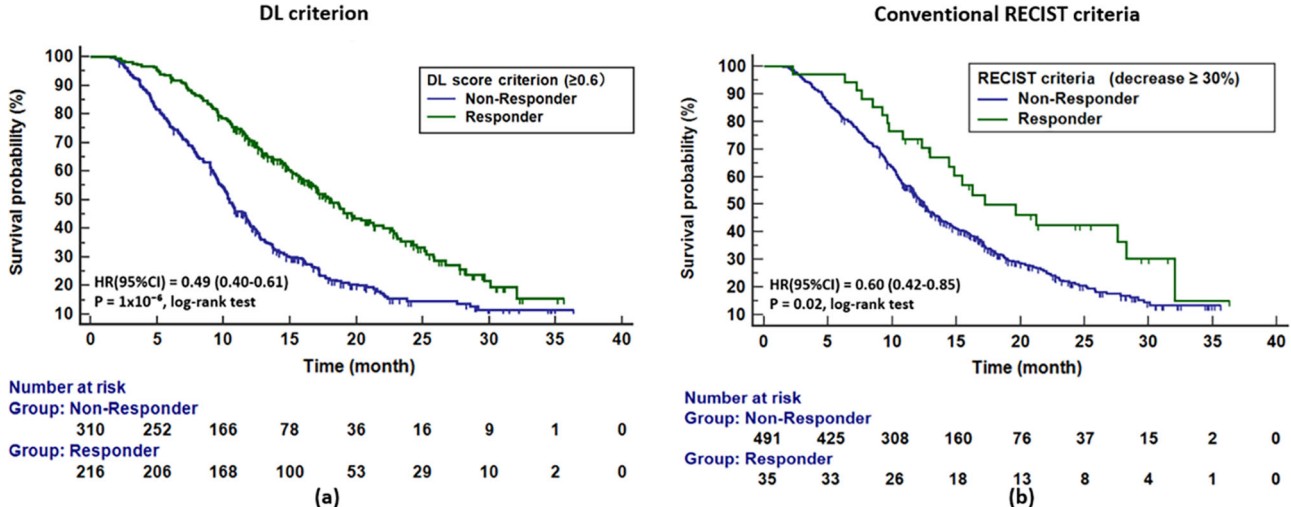

**Fig. 3 Survival analysis on DL score criterion and conventional RECIST criteria.** Plots of Kaplan-Meier estimator as well as hazard ratios and *p*-values estimated via log-rank test are present. For (**a**) DL score criterion, the responders are defined as score ≥ 0.6 and (**b**) for the conventional RECIST criteria, the responders are defined as tumor burden decreasing 30% or more.

| Table 3 Seven different prognostic models for early prediction of OS in patients with mCRC. | | | |
|---|---|---|---|
| **Model name (short name)** | **Cox proportional hazards model** | **Predictors (data type)** | **Time points used** |
| RECIST | Univariate | RECIST criteria (dichotomous) | Baseline and the 1st follow-up |
| Tumor Burden (TB) | Univariate | Measurement of TB (continuous) | Baseline |
| Early Tumor Shrinkage (ETS) | Univariate | Measurement of TB (continuous) | Baseline and the 1st follow-up |
| DL Baseline Score (DL-BS) | Univariate | Prediction score by DL network (continuous) | Baseline |
| DL Prediction Score (DL-PS) | Univariate | Prediction score by DL network (continuous) | Baseline and the 1st follow-up |
| Size Nomogram (Size-Nomo) | Multivariate | RECIST + TB + ETS | Baseline and the 1st follow-up |
| DL Nomogram (DL-Nomo) | Multivariate | DL-PS + RECIST + TB + ETS | Baseline and the 1st follow-up |

Note: '+' In the table indicates combination of predictors.

whom had similar tumor growth/shrinkage patterns but different OSs. In Fig. 6a, b, the tumor burden increased in both patients (all with 29% increase at month-2 follow-up), but patient (a) survived 8.2 months longer than patient (b). In Fig. 6a, activation mapping showed that, DL network paid attention to the occurrence of low-attenuation regions along tumor boundary (lesions #1, #2, and #4), indicating that such morphological patterns might lead to an improved prediction of outcomes. In Fig. 6c, d, the tumor burden decreased in both patients (one with 23% and the other with 22% decrease at month-2), but patient (d) survived 4.3 months less than patient (c). Interesting, in Fig. 6d, although tumors demonstrated decrease in size, the heterogeneity pattern (lesion #1) and vessel involvement (lesion #2) occurred at the month-2, which might suggest a prediction of short survival.

## Discussion

In this study, a DL network was constructed to predict early on-treatment OS in patients with mCRC using CT images retrieved at baseline and the first evaluation at 2 months. The validation in the test cohort including 526 mCRC patients demonstrated that patients with 'DL Score ≥ 0.6' predicted a prolonged survival with high probability. When compared with size-based ETS criteria, the DL-based criteria showed an enhanced ability to stratify patients into prolonged or short survivors. When incorporating DL prediction score with size-based predictors, the combined model, the DL-Nomo, achieved better prediction performance than any other size-based/DL-based model.

The motivation of this study was based in part on the understanding that size-based criteria have limitations for assessing

tumor response to anti-cancer therapies because size change only represents one dimension of tumor morphological change and does not include information on changes such as tumor density, heterogeneity, border thickness, etc. In clinical practice, different treatments might induce different tumor change patterns in mCRC patients, which may be difficult for size-based criteria alone to discern the complexities and differences.

Therefore, in this study, a DL network was proposed for characterization of tumor morphological change. The proposed DL network utilized two types of networks, the CNN and RNN. CNN is basically the type of network that is specified for image analysis and computer vision. Since the wide dissemination of CNN in the medical field over the past five years, it has already demonstrated remarkable capabilities in medical image classification[14–16], detection[17,18] and segmentation[30,31]. In this work, we followed the 'pretraining and fine-tuning' paradigm to transfer the GoogLeNet, which was pretrained on over 14 million images, from the domain of natural images to that of the mCRC CT images. The fine-tuned GoogLeNet showed promise on extracting morphological image features that were able to characterize tumor change beyond tumor size change (see Fig. 6). Although there was a correlation between CNN features and tumor size change, the correlation was weak (see Supplementary Fig. 5). Thus, it is reasonable that the integration of DL network, which was based on CNN features, could benefit the size-based methods by providing additional information on morphological change (see Fig. 5). Also, we adopted RNN to build a time-dependent model. RNN is basically the type of network designed for doing prediction based on temporal sequence. RNN allowed

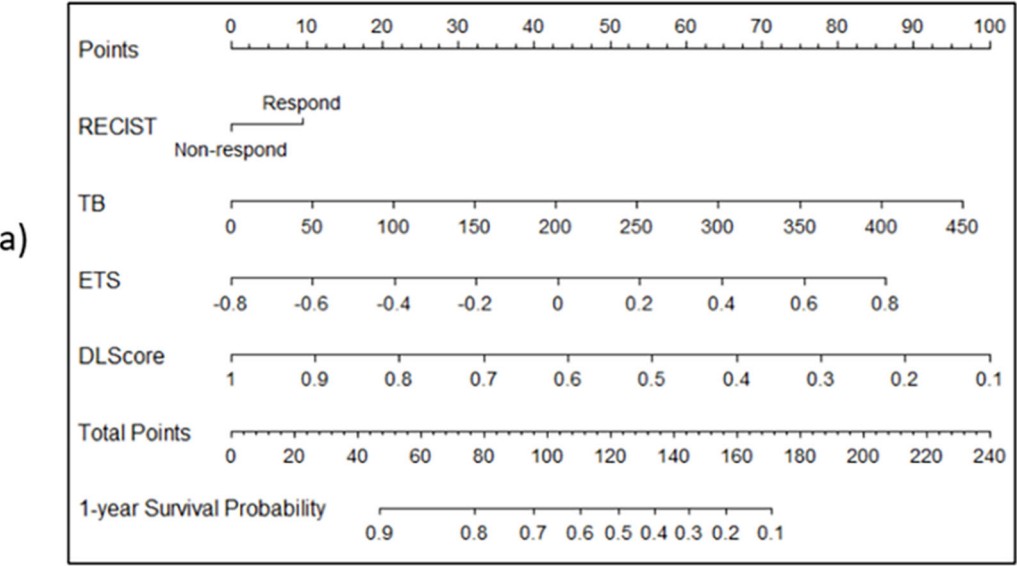

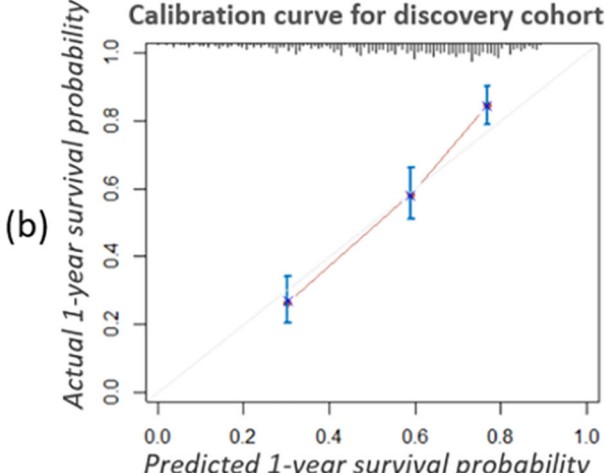

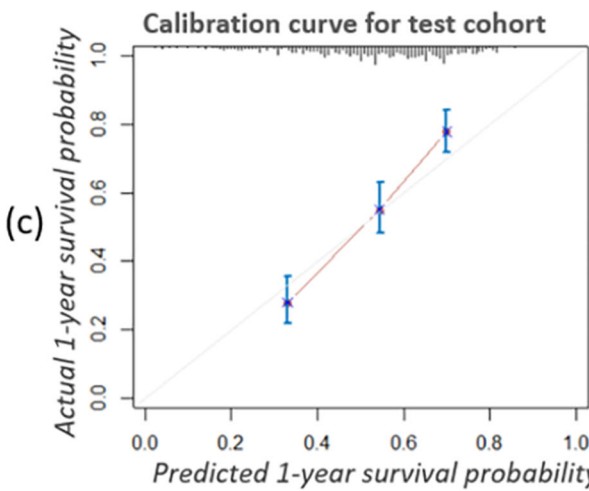

**Fig. 4 The nomogram for the DL-Nomo model with the assessment of its calibration. a** The nomogram for the DL-Nomo model. To use the nomogram, firstly, locate the patient's DL prediction score on the DLScore axis. Then, draw a line straight upward to the points axis to determine how many points toward the 1-year survival probability that the patient is received according to his or her DL prediction score. Repeat the same process for each variate. Thirdly, sum the points received from each of the variates and locate the final sum on the total point axis. Finally, draw a line straight down from the total point axis to the 1-year survival probability axis to find the patient's final 1-year survival probability. **b, c** The calibration curves for the DL-Nomo model in the discovery ($n = 502$) and test cohorts ($n = 526$), respectively. The calibration curve shows the calibration in terms of the agreement between the predicted and observed 1-year survival probability. Model-predicted 1-year survival probability is plotted on the *x*-axis, while the observed actual 1-year survival probability is plotted on the *y*-axis. The diagonal gray line indicates a perfect prediction by an ideal model, in which the predicted outcome perfectly corresponds to the actual outcome. The red lines indicate the performance of the model, a closer lining of which with the diagonal gray line represents a better estimation. And the predicted survival probability for the three patient groups (The short, median, and long survival groups with patients of $n =$ total patients/3 in each group) were shown as error bars, i.e., mean value ± standard error.

automated learning of time-dependent relation between features rather than using human curate modeling, e.g., the tumor growth inhibition modeling[32].

In addition to the application of DL method, strengths of our study also include the use of the VELOUR trial. First, VELOUR trial is an international, prospective, randomized study, allowing any results derived from it to be potentially incorporated into international guidelines in near future. Second, imaging data in the VELOUR trial are CT images. Although in recent years, imaging features derived from other imaging modalities (e.g., perfusion CT[33], contrast-enhanced ultrasound[34], diffusion–perfusion MRI[35], etc.) have been proposed as imaging biomarker of early on-

treatment prediction of patient outcome, however, these techniques have failed widespread implementation primarily because they are not the standard imaging techniques currently recommended by international guidelines for follow-up due to the lack of reproducibility. The DL-based criteria developed based on CT image has quite potential to be a widespread technique in both clinical trials and clinical practice. Third, the VELOUR trial included a total 1028 mCRC patients and 3757 lesions deriving from up to 14 anatomic locations, which was a large dataset in terms of cancer imaging, allowing development of DL method which favors large data input. Finally, the VELOUR trial included a total of 3193 CT scans, which were of great heterogeneity and wide coverage in terms of image

**Table 4 Harrell C-Index for the seven prognostic models.**

| Model name | Discovery cohort | | | Test cohort | | |
|---|---|---|---|---|---|---|
| | C-Index | 95% CI | *p*-value* | C-Index | 95% CI | *p*-value* |
| RECIST | 0.669 | 0.544,0.794 | 0 | 0.657 | 0.546,0.769 | 0 |
| TB | 0.621 | 0.590,0.652 | $1.7 \times 10^{-10}$ | 0.628 | 0.597,0.658 | $1.7 \times 10^{-5}$ |
| ETS | 0.639 | 0.608,0.670 | $7.9 \times 10^{-7}$ | 0.627 | 0.567,0.638 | $1.7 \times 10^{-5}$ |
| DL-BS | 0.618 | 0.588,0.648 | $6.5 \times 10^{-8}$ | 0.607 | 0.574,0.639 | $3.6 \times 10^{-10}$ |
| DL-PS | 0.678 | 0.650,0.706 | $1.5 \times 10^{-4}$ | 0.649 | 0.619,0.679 | $1.6 \times 10^{-4}$ |
| Size-Nomo | 0.677 | 0.649,0.707 | $8.9 \times 10^{-5}$ | 0.674 | 0.644,0.704 | $3.7 \times 10^{-3}$ |
| DL-Nomo | 0.707 | 0.680,0.734 | – | 0.694 | 0.661,0.720 | – |

Note: * Indicates the comparison to the DL-Nomo model. The *p*-values were calculated via two-sided *z*-test.

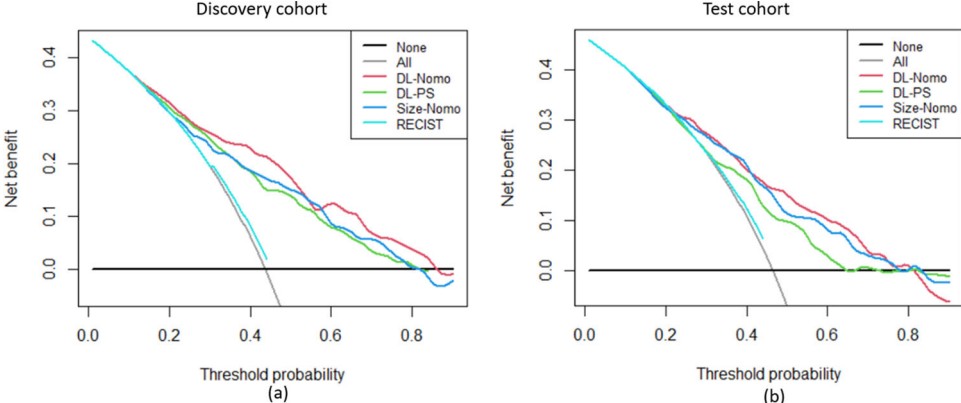

**Fig. 5 Decision curve analysis for the RECIST, Size-Nomo, DL-PS, and DL-Nomo models in the discovery (a) and test (b) cohorts, respectively.** In the figures, the *y*-axis measures the model benefit, and the *x*-axis shows the corresponding threshold. The model benefit is defined as the percentage of patients whose 1-year survival status (alive/dead) are correctly predicted. And the threshold is the cutoff used by the model, which is with the range of zero to one. As shown in the figures, the DL-Nomo model shows the highest level of benefit across the range of thresholds in both the discovery (**a**) and test (**b**) cohorts.

acquisition settings, allowing the development of a robustness algorithm for potential clinical use in practice.

As mentioned in the introduction, our work built upon the preliminary study of Y. Xu et al.[22] which utilized DL method to predict treatment response in lung cancer. However, in Y. Xu's work, the developed DL network did not show superior performance to size criteria. In contrary, our work demonstrated that DL network was capable to characterize tumor morphological change beyond size change. Some of the advantages of our DL method over the previous work could be, the use of more sophisticated network (GoogLeNet-Inception-v3 fine-tuned on tissue images), well-customized parameters for image pre-processing, and large number of data collected from well-organized clinical trial for network training. Another study that inspired our work was the study proposed by Dohan et al.,[9] which used a radiomics signature consisting of three image features to do treatment outcome prediction in patients with colorectal cancer. Although the radiomics signature showed good performance in the corresponding datasets, it still has the common limitations existing in radiomics approach, such as the manual/semi-manual contouring of lesion and the human engineering of image features.

Our study has several limitations. First, as a proof-of-concept study, we only included patients from a single clinical trial, although the trial was multicenter and did have two distinct trial arms. In future, as the developing of the Vol-PACT project[21], more patients from additional trials will be continuously collected to further validate the effectiveness of the DL-based criteria. For instance, recently, another two large-scale clinical trials on

mCRC, the CRYSTAL[36] (NCT00154102) and the PRIME (NCT04549935)[37], both of which contain more than one thousand patients, have already been available in the Vol-PACT. Secondly, the proposed DL criteria only utilized medical imaging information. In future, clinical information (e.g., patient characteristics) as well as other domain knowledge (e.g., genomics) will be incorporated into the DL network to further improve the performance. Finally, the interpretability of DL network is still limited, because the theory behind how the hidden neural units and layers interact and function has not yet been established[13]. Nevertheless, we believe that this limitation can be overcome in future as the development of DL theory[38,39].

In conclusion, this study demonstrated the impact of the proposed DL network on quantitative characterization of tumor morphological changes from pretreatment and follow-up CT scans which have association with patient's OS. There were increases in performance of OS prediction by using the proposed DL network. Since all required inputs to the DL network were only standard-of-care CT images and tumor measurement based on RECIST 1.1, it is hopeful that the DL network could be integrated into clinical practice as a computational tool to provide comprehensive and quantitative information on tumor morphological change for early on-treatment decision making at minimal or no cost.

## Methods

**Study design**. The design of our study is presented in Fig. 1. First, we retrieved data from the VELOUR trial from the Vol-PACT (Advanced Metrics and Modeling with Volumetric CT for Precision Analysis of Clinical Trial Results)[21]. A total of 1028

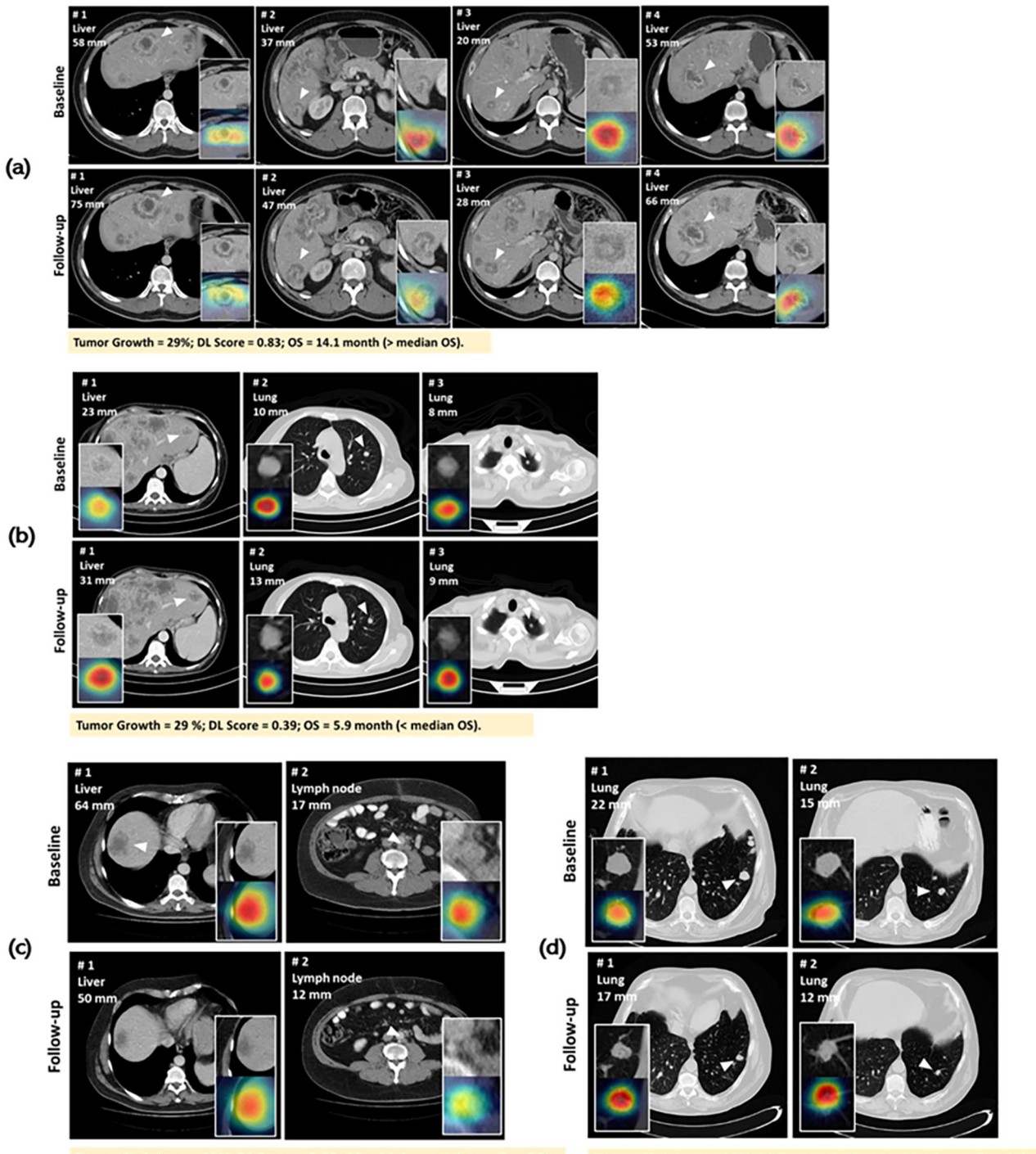

**Fig. 6 Examples for attention mapping on four mCRC patients.** In each picture, descriptions at the top-left corner show lesion's numbering, site and size measurement, white arrows indicate lesion location, and white frames at the bottom-left/right corner show the zooming region of interest and the correspond attention mapping for the region of interest. Each picture contained two rows, of which the upper showed CT scans of baseline and the bottom showed CT scans of follow-up. Descriptions in yellow at the bottom of each picture show the tumor shrinkage, deep learning (DL) method predicting score and overall survival (OS) for the correspond patients. **a**, **b** Were patients with tumor burden increasing, while (**c**, **d**) were patients with tumor burden decreasing; and although patients showed similar tumor load growth/shrinkage patterns, they had varying survival times.

patients with both OS data and CT images at baseline and month-2 were included. Patients were randomized into discovery and test cohorts, which were used to train and validate the DL networks, respectively. The randomization was provided by Vol-PACT along with the data. The analysis of our study consisted of two phases, the training and validation phases. In the training phase, the data in the discovery cohort were further randomized into training and tuning datasets, which were used to train network parameters and select optimal hyperparameters for the DL network, respectively. In the validation phase, the prognostic performance of DL network as

well as its incremental value to traditional size-based method were assessed. In addition, activation mapping technique was applied to provide visualization on region of interest defined by DL network to further human understanding.

**Datasets**. The VELOUR trial is an international prospective multi-institutional study that evaluated whether the antiangiogenic agent aflibercept in combination with 5-fluorouracil, leucovorin and irinotecan (FOLFIRI) could significantly

improve survival in a phase III study of patients with mCRC previously treated with an oxaliplatin-based regimen[20,24]. Detailed information of the VELOUR trial, including ethical regulation, trail approval, trial design, patient eligibility, patient randomization, and dose administration, can be found in the previous report[20]. The dataset from this trial provided for research by Vol-PACT[21] included 1141 mCRC patients with available OS data. After excluding patients without CT scans at month-2, a total of 1028 patients were used to create the discovery ($n = 502$ patients) and test cohorts ($n = 526$ patients) according to the randomization defined by the Vol-PACT. The discovery cohort was further split randomly into training and tuning sets at a ratio of 3:2. Training set was used to establish DL network, while tuning set was used to optimize network hyperparameter. Within each patient, up to 10 annotated 'target' lesions (metastases)[40] from whole body, whose diameters were greater than 1 cm at baseline, were used.

**The proposed DL networks.** The DL method used in this study deployed two types of networks, the convolutional neural network (CNN)[13] and the recurrent neural network (RNN)[41], as shown in Fig. 2. CNNs are typically used for automated extraction of image features from static images (e.g., image classification[42] and detection[43]), while RNNs are used to automatically build relation between dynamic time-dependent features from series of signals (e.g., speech recognition[44] and video analysis[45]). In our study of automated response assessment, CNN was utilized to extract image features from CT scan at each time point, while RNN was utilized to build up a time-dependent network from the image features extracted by CNN at the series of time points.

Our CNN architecture was the GoogLeNet (version Inception-v3[42], see Supplementary Fig. 9), initially pretrained on the ImageNet database[46] containing over 14 million natural images and then fine-tuned to mCRC image domain by using transfer learning according to a 'pretraining and fine-tuning' paradigm[47] (Details on fine-tuning the GoogLeNet were provided in 'GoogLeNet Fine-tuning' in the Supplementary Methods.). The proposed RNN architecture was composed of a series of bi-directional long short-term memory (LSTM)[41]. Bi-LSTM is an extension of traditional LSTM that can access information from past (backwards) and future (forwards) time points simultaneously. Image features extracted by the CNN at each time point (i.e., a vector consisting of 1024 features output by the last convolutional layer of GoogLeNet) were provided as input for the RNN training. Then, when RNN received inputs from CNN, it was trained to build an optimal combination of inputs to predict the patient outcome. The probability outputted by the softmax layer in the RNN is the final DL prediction score, that is, the DL-based signature to predict patient's probability to be a DL-responder/non-responder, i.e., the probability to be a prolonged/short survivor who survived longer/shorter than the median OS (In the VELOUR trail, the median OS is 12 months). The DL prediction score was a continuous value with a range from 0 to 1, where low value indicated poor survival and high value indicated good survival.

In the implementation, there points should be noted when training the CNN-RNN network. First, four image preprocessing procedures were applied to prepare inputs to the network as so to alleviate the impact of variation from CT image acquisition settings and selection of region of interest (ROI)[48]. They were; (1) normalize all CT images to homogenous voxel spacing of $1.0 \times 1.0 \times 1.0$ mm³ by using trilinear interpolation; (2) normalize image intensity a range of 0~255 by using CT window-levels; (3) crop lesion ROI from CT image, which was defined as a box of size $2d \times 2d$ (where $d$ is the length of diameter measured at baseline according to the RECIST 1.1 evaluation) with the center point corresponding to the center of the measurement line at baseline CT scan (It is noted that, lesion ROI at follow-up scan used the same $d$ as that at baseline scan); and (4) Spatially augment ROIs by applying random rotation ($-30°$~$30°$), shifting ($-0.05d$~$0.05d$), and scaling ($0.95d$~$1.05d$) (More details for the preprocessing were provided in 'ROI preparation' in the Supplementary Methods. Specially, a picture example for input of DL network on training was provided as Supplementary Fig. 10.). Subsequently, since most patients contained multiple targeted lesions, a weighted aggregation approach was used to combine the multi-lesion features, i.e., feature vectors extracted from each lesion in the same patient were element-wisely summarized into one vector by weighting with lesion diameters[10]. Third, when training RNN on the training set, the input number of time points was not limited, i.e., the number of follow-ups were allowed as many as possible, so as to reduce time-dependent signal noise. Of course, when testing on the test cohort, the input to RNN was limited by two, that is, only the baseline and the first follow-up at month-2 were allowed (Details on training the RNN were provided in 'RNN construction and training' in the Supplementary Methods.).

**Validation of DL prediction score.** The validation of prognostic performance of DL prediction score consisted of four steps. First, reproducibility analysis was performed to test the reproducibility of DL prediction score under potential variation of ROI selection (Details are provided in 'Reproducibility analysis' in the Supplementary Methods.). Second, in the tuning set, a cutoff for DL prediction score to stratify patients into high- or low-risk groups was determined according to the Youden Index[26] on the receiver operator characteristic curve (ROC), i.e., to select the point that was with the maximal value of sensitivity + specificity − 1 on ROC. Third, in the test set, Kaplan-Meier survival analysis was performed to assess the association between DL-based stratification (using the cutoff determined in the

tuning set) and patient's OS. Finally, the DL prediction score was compared to its size-based counterparts, the RECIST and ETS criteria.

**Assessment of incremental value of DL-based method.** To demonstrate the incremental value of DL-based method to traditional size-based method, the DL prediction score model as well as six other prognostic models were built via the Cox proportional hazards regression, and compared in terms of calibration, discrimination, and clinical usefulness. The seven models are listed in Table 3. Among the seven models, the RECIST, TB, ETS, and Size-Nomo models were traditional size-based methods, the DL-BS and DL-PS models were DL-based methods, and the DL-Nomo was the combination of the size- and DL-based methods.

To evaluate the calibration, calibration curves[27] were generated to measure the agreement between the observed outcomes and the model-predicted outcomes. To quantify the discrimination performance, Harrell C-Index (C-Index)[28] was employed. The Harrell C-Index was a commonly used algorithm to evaluate prognostic models in survival analysis, where data may be censored. For the Harrell C-Index, value 1.0 indicated perfect concordance, while value 0.5 indicated no better concordance than random choice. The decision curve analysis[29] was performed to evaluate the clinical usefulness of prognostic models by quantifying the net benefits at different threshold probabilities, that is, to see whether more net benefits could be attained by the addition of DL-based method to the tradition size-based methods.

**Network visualization.** Techniques of attention mapping could be used for indication of regions where DL network pay attention to[49,50]. In our work, the gradient-weighted activation mapping method (Grad-CAM)[23] was adopted to highlight the regions in an input CT image with respect to their contribution to the prediction by the GoogLeNet. The code of Grad-CAM was downloaded at https://github.com/ramprs/grad-cam/.

**Statistical analysis.** Statistical analyses were performed in MATLAB version 9.5, R version 4.0.1, MedCalc version 15.8, and Microsoft Excel 2019. In the reproducibility analysis, CCC[25] were used to evaluate the reproducibility of DL prediction score. In Kaplan-Meier analysis, log-rank test was used to compare the difference in the survival curves of high- and low-risk groups. The Cox proportional hazards regression was used to build prognostic model and to estimate the HR for predictors. The 95% confidence interval (CI) was provided as well. A landmark analysis correcting for bias was also performed. The comparison of two Harrell C-Indexes was implemented by R-package 'compareC'[51], which used nonparametric approach to estimate the variance of C estimators as well as their covariance and compared the two C-Indexes with a z-test. The calibration curves for models were plotted via R-package 'rms' (https://github.com/harrelfe/rms). Linear correlation between variables were indicated by correlation coefficient and $R^2$. The two-sample $t$-test and chi-square test were used to compare the difference within continuous and categorical data, respectively. A $p$-value less than 0.05 was used to indicate statistical significance, and all the statistical tests in this study were two sided.

**Reporting summary.** Further information on research design is available in the Nature Research Reporting Summary linked to this article.

## Data availability

The data that support the findings of this study are provided by the Vol-PACT (Advanced Metrics and Modeling with Volumetric Computed Tomography for Precision Analysis of Clinical Trial Results), which is a program under the Foundation for the National Institutes of Health (FNIH) Biomarkers Consortium (https://fnih.org/our-programs/biomarkers-consortium/programs/vol-pact). Since the fNIH Biomarkers Consortium is a public–private partnership, the data are available with permission from the Foundation's Biomarkers Consortium. Please contact the corresponding author of this paper who is also the PI of the Vol-PACT consortium to request access to the data. Data may be accessed by informing of the use for the data and permission of the consortium will be granted through the PI and project manager. Replies to initial requests will be made within 1 week and follow-up based upon the answers will be made within one consortium review cycle which is generally 3 months. While institutional membership is not obligatory for data sharing, an institution may also join the Vol-PACT Consortium by completing the application for participating the Vol-PACT through the program manager of Vol-PACT at the Biomarkers Consortium, Dana E. Connors (Senior Scientific Program Manager, Cancer Research Partnerships) at dconnors@fnih.org.

## Materials availability

L.S. is the guarantor of the present work, to whom correspondence should be addressed.

## Code availability

The complete source codes that support the findings of this study consist of three parts, data management, CNN feature extraction, and RNN construction and training. Among

them, the source codes for RNN construction and training in MATLAB language is publicly available at, https://github.com/LinLu1912/CNN-RNN-paper.git, as they do not involve direct access to the data provided by the Vol-PACT, which is a program under the FNIH Biomarkers Consortium (https://fnih.org/our-programs/biomarkers-consortium/programs/vol-pact). In contrary, the source codes for data management and CNN feature extraction are not publicly available because they involve direct access to the data provided by the Vol-PACT. To apply for the access of data as well as codes for data management and CNN feature extraction, please see the Data availability statement above.

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

## Acknowledgements
Authors acknowledge financial supports from the National Cancer Institute (U01 CA225431, LS and BZ). Authors acknowledge Scientific and financial support for the Foundation for the National Institutes of Health Biomarkers Consortium project Vol-PACT (Advanced Metrics and Modeling with Volumetric Computed Tomography for Precision Analysis of Clinical Trial Results) was provided by: Amgen; Boehringer Ingelheim; Merck KGaA, Darmstadt, Germany; Genentech Inc.; Merck & Co., Inc.; Regeneron Pharmaceuticals; and Millennium Pharmaceuticals, Inc., Cambridge, MA, a wholly owned subsidiary of Takeda Pharmaceutical Company Limited. In-kind donations of phase III trial data to support this specific study were provided to Foundation for the National Institutes of Health by Amgen and Sanofi. Authors acknowledge Dr. Tavis L. Allison's contribution on writing editing.

## Author contributions
L.L. designed the study, conducted image and statistical analysis, and drafted the manuscript. L.D. collected the data and edited the manuscript. B.S. and L.S. supervised the present study, edited, and approved the manuscript.

## Competing interests
The authors declare no competing interests.
