## [Peer Review File. · Nature Communications]

Reviewers' Comments:

Reviewer #1:

Remarks to the Author:

This study is a good application of using deep learning to predict early on-treatment response in metastatic colorectal cancer. The rich experiments illustrate the advantage of DL. However, the method and the whole pipeline is similar to current radiomics/DL studies, except that this is a new application. Although the DL score showed better performance than the commonly used ETS criteria, DL method may not be robust in practice. Most of the currently used criteria, such as ETS and RECIST, are very simple and easy to use, which make them robust between different centers. Radiomics/DL methods can probably show good results in a specific dataset, but they are sensitive to manual annotation (the box ROI selection) and CT scanning parameters.

1) This is a single dataset study, whether the DL model is robust among different centers needs further validation. Although the patients and CT protocols are collected from many centers, the random split of the training/testing sets make the training/testing sets have the same distribution in terms of follow-up and CT protocols. Namely, the DL model has seen part of the data distribution from all the involved centers. This probably lead to good results. However, when applying the DL model in a completely different center that is not seen by the model, the CT scanning parameters can probably affect the results. It's important to validate the robustness of the DL model in CT images with various scanning parameters.

2) This is still the concern about the robustness of the DL model in clinical practice. Since the authors annotated all the tumor-ROI for analysis, how will the results change if some of the ROI is missed? This can happen when using this method in clinical practice.

3) When applying this method, different operators can generate different ROI. The size and location of the ROI can change if it is annotated by different user, how will this inter-operator variance affect the results?

4) What is the performance of only use the baseline CT image? Please clarify the added value of using multiple CT sequences of different time.

5) Figure 5 is a good example. However, it's easy to find some examples that have fancy attention maps and the DL shows better performance than the commonly used clinical factors. Is it possible to provide a quantified measurement to support the conclusion in Fig.5? For example, how many cases can satisfy this conclusion?

Reviewer #2:

Remarks to the Author:

This paper entitled « Deep learning on predicting early on-treatment response in metastatic colorectal cancer from serial medical imaging” is very interesting.

However, I see several drawbacks in its present form.

A lot of studies in mCRC using DL have been published to date, and the novelty of the present study is not very clear to me. However, the cohort is relatively huge, and the DL techniques are deeply explained by the authors. However, the paper remains hard to read, and not very clear in the details.

Abstract: I am not used of this kind of abstracts with no results

Introduction:

Introduction id ok but the primary objective is not clear to me here. “to explore the ability of DL method to predict overall survival 92 (OS) in mCRC patients receiving...”

DL on which organ?

Method:

The DL part is very detailed.

The segmentation process and images extraction technique are not detailed at all. The radiological part of this paper should be seriously expanded

"Benchmarking against early tumor shrinkage" part: I d not understand why authors say they perfomed systematic review here?

The statistical analysis part is really limited, while there is a lot to explain here. How were cut off chosen ? how was performed internal validation ? a was p values adjusted to avoid overfitting ? How Kaplan Meier curves compared ?

Results:

Table 1 is unreadable

The effect of size response does not seem to have been taken into account while it is the strongest biomarker of survival. The influence of size response on other imaging feature should also be investigated

Reviewer #3:

Remarks to the Author:

In this paper, the authors have used data from the VELOUR trial to calculate a survival prognostic index for patients with metastatic colorectal cancer treated with FOLFIRI +/- aflibercept, using deep leaning (DL) techniques on serial imaging data available in the Vol-PACT database.

They compare the performance of their DL score to Early Tumor Shrinkage (ETS) and conclude that the DL-score has better operating characteristics than ETS. Although the results of this paper are extremely interesting, the paper is not clearly written, and the authors ignore important issues that need to be addressed before use of a DL-score becomes routine clinical practice.

First, prognostic scores using sophisticated techniques such as DL need to be assessed in terms of their additional benefit relative to simple prognostic scores based on easily available clinical and pathological features familiar to clinicians. If interest focuses on predicting OS, one would ideally like to know:

1. What patient and tumor factors can reliably predict OS
2. How much objective response assessed using RECIST adds to OS prediction 1
3. How much ETS adds to OS prediction 2
4. How much DL adds to OS prediction 3

In this paper one jumps right away to the last step, but without addressing the previous steps, the actual added value of the DL score remains unclear, however impressive the OS prediction appears to be.

Now the question of real interest is not OS prediction. It is the potential value of the DL score as a surrogate endpoint for future trials. Validation of the DL score as a surrogate endpoint for OS in mCRC would be a major breakthrough, but this would require access to several trials and assessment of the trial-level surrogacy, which is beyond the scope of the present paper. The authors, however, could mention this as a longer-term objective worthy of further investigation.

Minor comments

Page 4, line 107: "Patients were divided into discovery and test cohorts": how? At random? Have the authors considered repeating the random split, to assess robustness of the results?

Page 4, Figure 1: the boxes called "Benchmark" and "Visualization" are visually nice but their meaning is not clear.

Page 5, line 128: "The discovery cohort was further randomized...": rephrase to read "The discovery cohort was further split at random..."

Page 6: Details on the CNN and RNN architectures could be relegated without loss to the Supplementary Material.

Page 7, Statistical Analysis : descriptions are fuzzy and misleading.

- "Optimal cutoff for model to stratify patient were determined by Youden Index [37]". I get the meaning, but the sentence is unclear.
- "Hazard ratio (HR) and significance value were estimated by log-rank test". The hazard ratio is not estimated by the logrank test, and the significance value (P-value) is not estimated at all.
- "A landmark analysis correcting for bias was also performed". What was the landmark chosen, and why was there bias?
- "Linear correlation between variables were indicated by correlation coefficient and R-Squared". The correlation coefficient can be calculated whether the association is linear or not.
- "A p-value less than 0.001 was used to indicate statistical significance". Why this low p-value. If to adjust for multiplicity, what drives the choice of 0.001?

Page 7, line 201: VELOUR, not VOLOUR

Page 7, line 214 : P-values should not be given for correlation coefficients, as the misleadingly suggest a stronger correlation than actually exists.

Table 1 : Response Rate, not Respond Rate

Reviewer #1:

For convenience, the comments/requests are numbered, and the responses are highlighted in blue color. Modifications in the revised manuscript are highlighted in yellow color.

This study is a good application of using deep learning to predict early on-treatment response in metastatic colorectal cancer. The rich experiments illustrate the advantage of DL. However, the method and the whole pipeline is similar to current radiomics/DL studies, except that this is a new application.

#1. Although the DL score showed better performance than the commonly used ETS criteria, DL method may not be robust in practice. Most of the currently used criteria, such as ETS and RECIST, are very simple and easy to use, which make them robust between different centers. Radiomics/DL methods can probably show good results in a specific dataset, but they are sensitive to manual annotation (the box ROI selection) and CT scanning parameters.

Response:

Thank you for this helpful comment. We agree, this is a novel, and extremely important application for radiomics/DL which is why we undertook this study. We agree that the robustness problem must be well addressed. First, the data in this example lends itself to analysis of robustness as it is from a Phase 3 multicenter clinical trial, so in fact it is quite heterogenous and representative of what data would look like globally. When designing the proposed method, potential problems on the variation of ROI selection and the impact of CT scanning parameters were in fact fully taken into consideration.

A. Addressing variation of ROI selection

To address the variation of ROI selection, two special image preprocessing procedures, *i.e.*, a) RECIST-based ROI selection and b) spatial augmentation, were applied.

An additional reproducibility study was added to verify that the DL prediction score was reproducible against the potential variation of ROIs. This is the “ultimate” measure of reproducibility for this study.

Some details are as follows,

a) RECIST-based ROI selection

In this study, we do not employ/develop any additional manual/automatic annotation algorithms to determine/select ROIs. The generation of ROIs for the DL method were totally based on the measurements according to the RECIST 1.1 evaluation, that is,

“Crop lesion ROI from CT image, which was defined as a box of size $2d \times 2d$ (where d is the length of diameter measured at baseline according to the RECIST 1.1 evaluation) with the center point corresponding to the center of the measurement line at baseline CT scan (It is noted that, lesion ROI at follow-up scan used the same d as that at baseline scan).” See line 177-181. A picture example is also provided in the supplementary file as Figure S1.

We purposely used the design for two reasons. First, as was mentioned, the RECIST criteria is the most widely used response criteria and it is very simple to use and has already been demonstrated to be robust across different centers.

Secondly, the aim of the proposed method is to provide information on tumor morphological changes for traditional methods which are usually based on tumor size change. We do not aim to replace the use of size-based evaluation with a DL-based evaluation. Therefore, the proposed DL method was designed that we could take advantage of the size-based measurement. To stress this point, in the revised manuscript, additional comparative experiments are added (See the new subsection *“Assessment of incremental value of DL-based method to traditional size-based method”* at line 203-218).

And the comparative results showed that, when incorporating DL prediction score with size-based predictors, the combined model could achieve the best prediction performance than any other size-based / DL-based model (See the comparative results in line 291-340).

b) Spatial augmentation

The spatial augmentation technique is a widely used technique in the field of deep learning that has been proven to be very useful for increasing a model’s robustness against potential spatial variation. Here is some two references:

[1] “Shorten, Connor, and Taghi M. Khoshgoftaar. "A survey on image data augmentation for deep learning." *Journal of Big Data* 6.1 (2019): 1-48.”

[2] “Data Augmentation Techniques in CNN using Tensorflow.” (<https://medium.com/yomedicalabs-innovation/data-augmentation-techniques-in-cnn-using-tensorflow-371ae43d5be9>)

c) Reproducibility study and the results

In the revised manuscript, additional reproducibility analyses were performed to test the reproducibility of DL prediction score with a potential variation of ROI selection. The result showed that, the correlation concordance coefficient (CCC) for DL prediction score was 0.901 (95%CI: 0.880,0.913), suggesting good reproducibility of the DL prediction score even with the variation of ROI selection.

See line 194-195, 266-267, and supplement S4.

B. Addressing variation of CT imaging settings

As we briefly noted above, the strength of our study is in fact the robustness of CT scan acquisition settings as we used a global multicenter clinical trial, the VELOUR trial. We also used well established image normalization procedures.

a) Image normalization

Two image normalization procedures, the image spatial normalization and the image intensity normalization, were applied as follows,

“1) normalize all CT images to homogenous voxel spacing of 1.0x1.0x1.0 mm³ by using tri-linear interpolation; 2) normalize image intensity a range of 0~255 by using CT window-levels.” See line 176-177.

These two image preprocessing procedures are usual approaches to harmonize spatial resolution and image display ranges for CT images acquired via different CT scanners.

b) The VELOUR data

In addition to the image preprocessing, the use of the VELOUR data is also very important. Since the VELOUR trial is an international multi-institutional study, the CT images we used are of great heterogeneity in imaging acquisition settings. In the revised manuscript, details for CT scanning characteristics are provided in the new Table 3. See line 263.

“As shown in Table 3, there were a total of 3,193 CT scans used, deriving from a total of 9 CT manufacturers, 81 manufacturer models and 65 CT image reconstruction algorithms, respectively. There was significant difference between the discovery and test cohorts in terms of manufacturer models and CT image reconstruction algorithms (both $p < 0.05$). For those regular imaging parameters, e.g., slice thickness, voltage, product of tube current and time, and pixel spacing, there was no significant difference (all $p > 0.05$ except the product of tube current and time) and mainstream ranges of settings were covered.” See line 254-260.

The great heterogeneity and wide coverage of VELOUR data allowed the development of a robust algorithm for potential wide, global clinical use in medical practice.

#2. This is a single dataset study, whether the DL model is robust among different centers needs further validation. Although the patients and CT protocols are collected from many centers, the random split of the training/testing sets make the training/testing sets have the same distribution in terms of follow-up and CT protocols. Namely, the DL model has seen part of the data distribution from all the involved centers. This probably lead to good results. However, when applying the DL model in a completely different center that is not seen by the model, the CT scanning parameters can probably affect the results. It’s important to validate the robustness of the DL model in CT images with various scanning parameters.

Response:

Thank you for this thoughtful comment. It is true that an algorithm developed based on single dataset is vulnerable to the variation of CT acquisition parameters as most of the data may be acquired by similar scanning machines and CT protocols. Therefore, in the revised manuscript, important CT acquisition parameters that are available in DICOM header were retrieved and provided in the new Table 3. See line 263.

As shown in Table 3, there were totally 3,193 CT scans used from the VELOUR data. Since the VELOUR data is an international, prospective, multi-national and randomized study, the 3,193 CT scans covered all of the four main CT-brands, GE, SIEMENS, Philips, TOSHIBA, which involved up to a total of 81 machine models that had significant different distributions between the discovery and test cohorts

($p < 0.05$). The 81 machine models used up to a total of 65 different reconstruction convolution kernels which also showed significant different distributions between the discovery and test cohorts ($p < 0.05$). A detail related to clinical trial acquisition is that most often the baseline study is not performed in the same center as the follow up study – i.e., the baseline study is prior to study entry and frequently at the time of progression from a prior treatment. Also, while many patients have follow-up studies performed at the center where they are being treated, many patients prefer the follow up scans closer to home at a different and unique location. Therefore, the extreme heterogeneity of this data as noted above in fact limits the ability for the “DL model has seen part of the data distribution from all the involved centers”.

The heterogeneity of VELOUR data is helpful for developing a robustness algorithm. One evidence is that the prognostic performance for the DL prediction score model only dropped a little when trained on discovery cohort (C-Index: 0.678 (95%CI: 0.650,0.706)) and then validated in the test cohort (C-Index: 0.649 (95%CI: 0.619,0.679)). See the new Table 4 in line 322.

#3. This is still the concern about the robustness of the DL model in clinical practice. Since the authors annotated all the tumor-ROI for analysis, how will the results change if some of the ROI is missed? This can happen when using this method in clinical practice.

Response:

As mentioned above. The tumor-ROI for analysis is generated according to the measurement of the RECIST 1.1 evaluation (See the “a) RECIST-based ROI selection” in the response to comment #1 above). So, if an ROI is missed, it is easy for radiologist to draw a measurement line on lesion and then re-generate the ROI. Of course, if an error is made in “interpretation” there will be a similar effect on the DL model as on the re RECIST calculation.

#4. When applying this method, different operators can generate different ROI. The size and location of the ROI can change if it is annotated by different user, how will this inter-operator variance affect the results?

Response:

See the “A. Addressing variation of ROI selection” in the response to comment #1 above.

#5. What is the performance of only use the baseline CT image? Please clarify the added value of using multiple CT sequences of different time.

Response:

In the revised manuscript, additional comparative experiments are added (See the new subsection “Assessment of incremental value of DL-based method to traditional size-based method” at line 203-218).

According to the new results shown in Table 4, the DL prediction score model using multiple time points (the DL-PS model) showed better performance than that of only using CT at baseline (the DL-BS model), with C-Index of 0.649 (95%CI:0.619,0.679) vs. 0.607 (95%CI:0.574, 0.639), $p < 0.05$. See line 316-318.

#6. Figure 5 is a good example. However, it's easy to find some examples that have fancy attention maps and the DL shows better performance than the commonly used clinical factors. Is it possible to provide a quantified measurement to support the conclusion in Fig.5? For example, how many cases can satisfy this conclusion?

Response:

Thank you for this thoughtful comment.

In the field of deep-learning based computer vision, there are techniques, such as CAM [1] and Grad-CAM [2], which attempt to generate "visual explanations" for decisions and help to establish appropriate trust in predictions. However, since deep-learning network is a 'black-box' usually containing millions of parameters, there is no reliable quantified measurement for the "visual explanations" until now. Of course, as the development of deep-learning methodology progresses, we believe the problem on quantifying "visual explanations" for network can be solved.

In the revised manuscript, we added this point as a limitation to the discussion section and also added references [3,4] on the development of quantifying visual explanations See line 426-429.

[1] Zhou, Bolei, et al. "Learning deep features for discriminative localization." Proceedings of the IEEE conference on computer vision and pattern recognition. 2016.

[2] Selvaraju, R.R., et al. Grad-cam: Visual explanations from deep networks via gradient-based localization. in Proceedings of the IEEE international conference on computer vision. 2017.

[3] Bau, D., et al., Understanding the role of individual units in a deep neural network. Proc Natl Acad Sci U S A, 2020. 117(48): p. 30071-30078.

[4] Fan, F.-L., et al., On interpretability of artificial neural networks: A survey. IEEE Transactions on Radiation and Plasma Medical Sciences, 2021: p. 1.

Reviewer #2:

For convenience, the comments/requests are numbered, and the responses are highlighted in blue color. Modifications in the revised manuscript are highlighted in yellow color.

This paper entitled "Deep learning on predicting early on-treatment response in metastatic colorectal cancer from serial medical imaging" is very interesting.

However, I see several drawbacks in its present form.

#1. A lot of studies in mCRC using DL have been published to date, and the novelty of the present study is not very clear to me. However, the cohort is relatively huge, and the DL techniques are deeply explained by the authors. However, the paper remains hard to read, and not very clear in the details.

Response:

Thanks for the comment. While studies have been published with mCRC, the use of serial imaging and DL for response assessment, as noted by the Reviewer #1, is indeed novel. With regards to the clarity of the paper, we completely agree and have modified the manuscript and added relevant content to the Introduction, Method, and Results sections to make the manuscript and the paper easier to read.

A. The Introduction section

In the Introduction section, the status as well as the challenges of using DL to predict mCRC treatment response are summarized and added to the revised manuscript as follows,

“Until now, there is few reports on applying DL method on predicting early on-treatment response in mCRC. For example, when searching in the National Library of Medicine medical literature database with the key words of ‘deep learning’, ‘metastatic colorectal cancer’ and ‘response or survival’ (<https://pubmed.ncbi.nlm.nih.gov/>), only one reference was identified [19] and the literature was only about colorectal liver metastases with small data. Based on our knowledge, the challenges on predicting mCRC treatment response by using DL method include, 1) mCRC involves multiple metastatic lesions located in different anatomic regions rather than a single organ; 2) the prediction of response involves CT images of multiple time points rather than a single time point; and 3) DL method requires large data set for training.” See line 92-101.

Thus, the novelty of the present study lies in the strength to address the challenges mentioned above.

“To address the challenges mentioned above, in our study, a sophisticated DL network architecture, which combines the convolutional neural network (CNN) [19] and the recurrent neural network (RNN) [35], were proposed. The CNN was used to characterize lesions from different anatomic positions, while the RNN was used to handle the characterizations of multiple time points. Also, a total of 1028 mCRC patients collected from the VELOUR trial (an international prospective multi-institutional study [20]) were used for our study.” See line 103-109.

B. The Method section

a) Details on preparing ROIs for the DL network

We have revised the manuscript with details of the four image preprocessing procedures for preparing ROIs for the DL network, including 1) image spatial normalization, 2) image intensity normalization, 3) determination of region of interest (ROI), and 4) spatial augmentation of ROIs. See line 176-183.

b) Assessment of incremental value of DL-based method to traditional size-based method

We have revised the manuscript with a new subsection to demonstrate the incremental value of DL based method to traditional size-based method. In this section, the DL prediction score model as well as six other prognostic models are built via the Cox proportional hazards regression, and compared in in terms of calibration, discrimination, and clinical usefulness. See line 203-218 for details.

Of note, the previous content on comparing DL prediction score and ETS, *i.e.*, the subsection “Benchmarking comparison”, has been moved to the supplementary file (S10).

C. The Result section

a) Lesion characteristics and CT scanning characteristics

The data we used include a total of 1,028 mCRC patients with up to 3,757 lesions and 3,193 CT scans. Now, in the expanded “Patient Characteristics” section, the details for characteristics of lesion and CT scanning are also provided in the new Table 2 and 3, respectively. See line 261-264.

b) Results for assessment of incremental value

As mentioned above, the DL prediction score model as well as six other prognostic models are built and compared. Thus, in the Result section, the corresponding results are provided. Overall, the combined model which incorporating both DL-based and size-based predictors achieved the best performance of all. See line 291-340.

#2. Abstract: I am not used of this kind of abstracts with no results

Response:

We have revised the manuscript with pertinent results and now added to the Abstract. See line 39-43.

Introduction:

#3. Introduction id ok but the primary objective is not clear to me here. “to explore the ability of DL method to predict overall survival 92 (OS) in mCRC patients receiving...”

DL on which organ?

Response:

The mCRC can involve multiple metastasis lesions deriving from different anatomic locations, which is one of the main challenges on analyzing mCRC by using computational method. In the revised manuscript, this point was mentioned in the Introduction (See line 98-99). mCRC can spread to any organ or tissue – this is the definition of metastatic disease – this is why this work is so important.

In the revised manuscript, to demonstrate this point, the lesion characteristics are also provided in Table 2. We can see that, the 1,028 patients we used totally involved 3,757 lesions (averagely 3.53 ± 2.19 target

lesions per patient) coming from a total of 14 anatomic locations. And the main lesion sources are liver (58.3%), lung (22.5%), and lymph node (12.0%). See line 248-250.

Method:

#4. The DL part is very detailed.

Response:

Thank you.

#5. The segmentation process and images extraction technique are not detailed at all. The radiological part of this paper should be seriously expanded

Response:

Thank you for your thoughtful suggestions. The image preprocessing procedures to prepare inputs of DL network are expanded. See line 176-183. And more details on image preprocessing, including a picture example, can be found in the supplementary file as supplement S2.

#6. “Benchmarking against early tumor shrinkage” part: I do not understand why authors say they performed systematic review here?

Response:

You are correct, and we have modified the manuscript with regards to the statement of “systematic review” which is overstated. The comparison between DL and ETS criteria is actually based on the existing literatures searched in the National Library of Medicine medical literature database. There are eight mCRC-related trials are found, and they all have reports on ETS evaluation. The eight trails are the BOND, 20050181 study, CRYSTAL, OPUS, PRIME, FIRE-3, PEAK, and TRIBE.

Now, in the revised manuscript, since such comparison could not directly show the incremental clinical value of DL prediction score, we moved it to the supplementary file as supplement S10 for the sake of readers’ interest. Instead, we conducted comparative experiments to demonstrate the incremental value of DL based method to traditional size-based method. See the new subsection “Assessment of incremental value of DL-based method to traditional size-based method” in line 203-218.

#7. The statistical analysis part is really limited, while there is a lot to explain here. How were cut off chosen? how was performed internal validation? was p values adjusted to avoid overfitting? How Kaplan Meier curves compared?

Response:

Thank you for your thoughtful suggestions. We have revised the manuscript, and the subsections, "Validation of Prognostic Performance of DL prediction score" and "Statistical Analysis" are greatly expanded and reorganized.

a) In the subsection of "Validation of Prognostic Performance of DL prediction", the way to choose cutoff and to perform internal validation are presented as follow,

"In the tuning set, a cutoff for DL prediction score to stratify patients into high-risk or low-risk groups was determined according to the Youden Index [34] on the receiver operator characteristic curve (ROC), i.e., to select the point that was with the maximal value of sensitivity+specificity-1 on ROC. In the test set, Kaplan-Meier survival analysis was performed to assess the association between DL-based stratification (using the cutoff determined in the tuning set) and patient's OS." See line 195-200.

b) For the p-value adjustment, since image features within DL network are automatically learned during the training process, there is no need to do adjustment for features.

c) In the subsection of "Statistical Analysis", the way to compare survival curves is presented as follow,

"In Kaplan-Meier analysis, log-rank test was used to compare the difference in the survival curves of high- and low-risk groups." See line 233-235.

Results:

#8. Table 1 is unreadable

Response:

We have revised the manuscript and the previous Table 1 has also been modified and was moved to the supplementary file as supplement Table S1.

#9. The effect of size response does not seem to have been taken into account while it is the strongest biomarker of survival. The influence of size response on other imaging feature should also be investigated

Response:

That is a very helpful suggestion.

As mentioned in the response to comment #1, now, a new subsection called "Assessment of incremental value of DL-based method to traditional size-based method" is added to provide details on demonstrating the incremental value of DL based method to traditional size-based method. See line 203-218. And the corresponding comparative results are provided in the Result section as well. See line 291-340.

Reviewer #3:

For convenience, the comments/requests are numbered, and the responses are highlighted in blue color. Modifications in the revised manuscript are highlighted in yellow color.

In this paper, the authors have used data from the VELOUR trial to calculate a survival prognostic index for patients with metastatic colorectal cancer treated with FOLFIRI +/- aflibercept, using deep learning (DL) techniques on serial imaging data available in the Vol-PACT database.

They compare the performance of their DL score to Early Tumor Shrinkage (ETS) and conclude that the DL-score has better operating characteristics than ETS. Although the results of this paper are extremely interesting, the paper is not clearly written, and the authors ignore important issues that need to be addressed before use of a DL-score becomes routine clinical practice.

#1. First, prognostic scores using sophisticated techniques such as DL need to be assessed in terms of their additional benefit relative to simple prognostic scores based on easily available clinical and pathological features familiar to clinicians. If interest focuses on predicting OS, one would ideally like to know:

1. What patient and tumor factors can reliably predict OS
2. How much objective response assessed using RECIST adds to OS prediction 1
3. How much ETS adds to OS prediction 2
4. How much DL adds to OS prediction 3

In this paper one jumps right away to the last step, but without addressing the previous steps, the actual added value of the DL score remains unclear, however impressive the OS prediction appears to be.

Response:

Thank you for this thoughtful analysis. We have modified the manuscript and added additional comparative experiments to demonstrate the incremental value of the DL-based method to traditional size-based methods. We did not analyze the incremental value of patient and tumor factors (e.g., age, sex, smoking status, genomic mutant status, etc.) as those were included in the clinical trial result papers, because in this paper we wanted to initially focus our attention on medical imaging. Ultimately, in future data sets we will combine all the available factors.

In the Method section, a new subsection “Assessment of incremental value of DL-based method to traditional size-based method” is added to provide details for the additional comparative experiments. See line 203-218 for details.

Briefly, “The DL prediction score model as wells as six other prognostic models were built via the Cox proportional hazards regression, and compared in terms of calibration, discrimination, and clinical usefulness.” For your convenience, the new Table 1 listing the seven prognostic models are pasted below.

Table 1. Seven different prognostic models for early prediction of OS in patients with mCRC.

Model Name (Short name)	Cox proportional hazards model	Predictors (Data Type)	Time Points Used
RECIST	univariate	RECIST criterion (dichotomous)	Baseline and the 1 st follow-up
Tumor Burden (TB)	univariate	Measurement of TB (continuous)	Baseline
Early Tumor Shrinkage (ETS)	univariate	Measurement of TB (continuous)	Baseline and the 1 st follow-up
DL Baseline Score (DL-BS)	univariate	Prediction score by DL network (continuous)	Baseline
DL Prediction Score (DL-PS)	univariate	Prediction score by DL network (continuous)	Baseline and the 1 st follow-up
Size Nomogram (Size-Nomo)	multivariate	RECIST + TB + ETS	Baseline and the 1 st follow-up
DL Nomogram (DL-Nomo)	multivariate	DL-PS + RECIST + TB + ETS	Baseline and the 1 st follow-up

Note: “+” in the table indicates combination of predictors.

As noted, the previous content on comparing DL prediction score and ETS, *i.e.*, the subsection “Benchmarking comparison”, has been moved to the supplementary file as supplement S10, because it does not directly show the incremental clinical value of DL prediction score.

In the Result section, the new subsection “Assessment of incremental value of DL-based method to traditional size-based method” is added to provide the results of these important comparative experiments. The results show that, the DL prediction score model (DL-PS) shows better performance than its size-based counterpart, the ETS model (C-Index: 0.649 (95%CI: 0.619,0.679) vs. 0.627 (95%CI: 0.567,0.638), $p < 0.05$), and the combination model, the DL-Nomo, shows the best prediction performance among all the comparative prognostic models. See line 312-340 for detail.

#2. Now the question of real interest is not OS prediction. It is the potential value of the DL score as a surrogate endpoint for future trials. Validation of the DL score as a surrogate endpoint for OS in mCRC would be a major breakthrough, but this would require access to several trials and assessment of the trial-level surrogacy, which is beyond the scope of the present paper. The authors, however, could mention this as a longer-term objective worthy of further investigation.

Response:

Thanks for this comment and we completely agree! The good news is that in the future, we believe that we will be able to accomplish this goal and the publication of this data will help us to achieve this. Our

consortium, Vol-PACT is gaining momentum and as a result other clinical trial data has been suggested and we are working out the details of the transfer. For instance, recently, another two large-scale clinical trials on mCRC, the CRYSTAL (NCT00154102) and the PRIME (NCT04549935), both of which contain more than one thousand patients, will be available in Vol-PACT. In near future, we hope to apply this technology to these new data sets.

The above information on new available data has been added to the Discussion section. See line 420-424.

Minor comments

#3. Page 4, line 107: “Patients were divided into discovery and test cohorts”: how? At random? Have the authors considered repeating the random split, to assess robustness of the results?

Response:

Yes. The Patients are randomized into discovery and test cohorts. The randomization is provided by the Vol-PACT along with the downloaded data. The use of such randomization enables un-biased comparison between current study and future study by other researchers. This information has been added. See line 123-124.

Also, in the revised manuscript, more details on the randomization are provided, including the distributions of lesion and CT scanning setting between the discovery and test cohorts. See line 248-250, line 254-260 and the new Table 2 and 3.

#4. Page 4, Figure 1: the boxes called “Benchmark” and “Visualization” are visually nice but their meaning is not clear.

Response:

Thanks for the suggestions. Now, the Figure 1 has been improved by adding more details to the boxes. See line 132.

It is noted that, the previous box “Benchmark” has been changed to the “Assessment of incremental value to traditional method” according to the revised Method.

#5. Page 5, line 128: “The discovery cohort was further randomized...”: rephrase to read “The discovery cohort was further split at random...”

Response:

Done. See line 141-142.

#6. Page 6: Details on the CNN and RNN architectures could be relegated without loss to the Supplementary Material.

Response:

The descriptions on architectures of CNN and RNN have been shortened and cleared up. In addition, more of the details were moved to the supplementary file as supplements S1-S3. And the supplementary file has also been improved to be read more clearly.

#7. Page 7, Statistical Analysis: descriptions are fuzzy and misleading.

#7-1. • “Optimal cutoff for model to stratify patient were determined by Youden Index [37]”. I get the meaning, but the sentence is unclear.

#7-2. • “Hazard ratio (HR) and significance value were estimated by log-rank test”. The hazard ratio is not estimated by the logrank test, and the significance value (P-value) is not estimated at all.

#7-3. • “A landmark analysis correcting for bias was also performed”. What was the landmark chosen, and why was there bias?

#7-4. • “Linear correlation between variables were indicated by correlation coefficient and R-Squared”. The correlation coefficient can be calculated whether the association is linear or not.

#7-5. • “A p-value less than 0.001 was used to indicate statistical significance”. Why this low p-value. If to adjust for multiplicity, what drives the choice of 0.001?

Response:

The section “Statistical Analysis” has been reorganized and improved.

#7-1. The sentence was expanded to make it clearer on how to select cutoff, as shown below.

“In the tuning set, a cutoff for DL prediction score to stratify patients into high-risk or low-risk groups was determined according to the Youden Index [34] on the receiver operator characteristic curve (ROC), i.e., to select the point that was with the maximal value of sensitivity+specificity-1 on ROC.” See line 195-198.

#7-2. Corrected. See below,

“In Kaplan-Meier analysis, log-rank test was used to compare the difference in the survival curves of high- and low-risk groups. The Cox proportional hazards regression was used to build prognostic model and to estimate Hazard ratio (HR) for predictors.” See line 233-235.

#7-3. The landmark is at month-2. Since the 1,028 patients included in this study should have at least two CT scans, i.e., the scanning at base line and 1st follow-up, that means, all of the patients must live longer than 2 months.

#7-4. The statement is removed. And the linear correlation evaluation between the DL score and ETS has been moved to the supplementary file as the supplement S8, because the comparative experiments

mentioned above (see the response to comment #1) is able to better demonstrate the benefit of DL method to the traditional size-based method.

#7-5. Checked and corrected. Now, the threshold of 0.05 for p-value is used consistently through the entire study. See line 239.

#8. Page 7, line 201: VELOUR, not VOLOUR

Response:

Corrected. See line 245.

#9. Page 7, line 214: P-values should not be given for correlation coefficients, as they misleadingly suggest a stronger correlation than actually exists.

Response:

Done. The p-value for correlation coefficient has been removed. See line 280-282.

#10. Table 1: Response Rate, not Respond Rate

Response:

See the Table S1 in the supplementary file please. The previous Table 1 has been moved to the supplementary file as supplement Table S1.

Reviewers' Comments:

Reviewer #1:

Remarks to the Author:

Many comparative experiments have been performed to clarify the robustness of the DL model regarding different CT imaging protocols. Details about the ROI annotation is also described. The added value of the multi-time point CT image over using the baseline CT image is also presented. The authors have clarified all my concerns.

Reviewer #2:

Remarks to the Author:

Most of the remarks have been adequately addressed.

The performances of the DL model are weaker than the SPECTRA score published elsewhere : Early evaluation using a radiomic signature of unresectable hepatic metastases to predict outcome in patients with colorectal cancer treated with FOLFIRI and bevacizumab Gut. 2020 Mar;69(3):531-539. doi: 10.1136/gutjnl-2018-316407. Epub 2019 May 17.

which is more simple. Finally does the DL add anything to the size and density criteria alone ? could authors comment on this ?

Reviewer #3:

Remarks to the Author:

The authors should be commended for their thorough review of the paper along the lines suggested in the reviewers' comments. A few further details would benefit from being fixed prior to publication.

Minor comments

Abstract, page 2 line 41: replace $P < 0.05$ by actual P-value. (See related comments below).

Abstract, page 2 line 43: replace $P < 0.05$ by actual P-value and specify what test this P-value corresponds to.

Figure 3, page 11: "Conventional RECIST criteria" instead of "Convention RECIST criterion"

Page 13, lines 315 and 317: replace $P < 0.05$ by actual P-values

Page 13, table 4: replace $P < 0.05$ by actual P-values

For your convenience, the comments/requests are numbered, and the responses are highlighted in blue color. Modifications in the revised manuscript are highlighted in yellow color.

Reviewer #1:

Many comparative experiments have been performed to clarify the robustness of the DL model regarding different CT imaging protocols. Details about the ROI annotation is also described. The added value of the multi-time points CT image over using the baseline CT image is also presented. The authors have clarified all my concerns.

Response:

Thank you very much.

Reviewer #2:

Most of the remarks have been adequately addressed.

The performances of the DL model are weaker than the SPECTRA score published elsewhere: Early evaluation using a radiomic signature of unresectable hepatic metastases to predict outcome in patients with colorectal cancer treated with FOLFIRI and bevacizumab Gut. 2020 Mar;69(3):531-539. doi: 10.1136/gutjnl-2018-316407. Epub 2019 May 17.
which is more simple.

Finally does the DL add anything to the size and density criteria alone?

Could authors comment on this?

#1. The performances of the DL model are weaker than the SPECTRA score published elsewhere: Early evaluation using a radiomic signature of unresectable hepatic metastases to predict outcome in patients with colorectal cancer treated with FOLFIRI and bevacizumab Gut. 2020 Mar;69(3):531-539. doi: 10.1136/gutjnl-2018-316407. Epub 2019 May 17.
which is more simple.

Response: Thanks a lot for the suggestion.

The SPECTRA-Score mentioned above is really a nice work on using computational method to do early treatment response assessment for colorectal cancer. Therefore, even in the original manuscript, the work of SPECTRA-Score had been cited as a reference in the Introduction section,

“Many qualitative and quantitative image analysis algorithms for assessing tumor morphological change have been proposed over the last decade [8-11].” (The reference [9] is the SPECTRA-Score paper)

So now, in the revised manuscript, as you suggested, a brief introduction (See line 85-88 in the Introduction section) as well as the corresponding comment (See line 255-260 in the Discussion section) were provided.

#2. Finally does the DL add anything to the size and density criteria alone?

Response:

Yes. As shown in Table 4, the final DL-Nomo model, which incorporated DL and Size information, showed better performance than the ETS model (size-based model, see Table 3 for its definition) and the DL-PS model (DL-based model, see Table 3 for its definition) on both discovery and test cohorts. This point has been mentioned in the Results section (See line 175-177) as well as the Discussion section (See line 221-228).

Reviewer #3:

The authors should be commended for their thorough review of the paper along the lines suggested in the reviewers' comments. A few further details would benefit from being fixed prior to publication.

Minor comments

#1. Abstract, page 2 line 41: replace $P < 0.05$ by actual P-value. (See related comments below).

Response: Done. See line 41.

#2. Abstract, page 2 line 43: replace $P < 0.05$ by actual P-value and specify what test this P-value corresponds to.

Response:

Done. See line 43.

In addition, we provided the implementation information in the Statistical Analysis section. For your convenience, the added contents are pasted below,

“The comparison of two Harrell C-indexes was implemented by R-package ‘compareC’ [51], which used nonparametric approach to estimate the variance of C estimators as well as their covariance and compared the two C indexes with a z test.” See line 394-396.

#3. Figure 3, page 11: “Conventional RECIST criteria” instead of “Convention RECIST criterion”

Response: Done. See the Revised Figure 3, please.

#4. Page 13, lines 315 and 317: replace $P < 0.05$ by actual P-values

Response: Done. See line 171 and 173.

#5. Page 13, table 4: replace $P < 0.05$ by actual P-values

Response: Done. See the Revised Table 4, please.